# Negative Energy Balance Influences Nutritional Quality of Milk from Czech Fleckvieh Cows due Changes in Proportion of Fatty Acids

**DOI:** 10.3390/ani10040563

**Published:** 2020-03-27

**Authors:** Jaromír Ducháček, Luděk Stádník, Martin Ptáček, Jan Beran, Monika Okrouhlá, Matúš Gašparík

**Affiliations:** 1Department of Animal Science, Faculty of Agrobiology, Food and Natural Resources, Czech University of Life Sciences, Kamýcká 129, Suchdol, 165 00 Prague 6, Czech Republic; stadnik@af.czu.cz (L.S.); ptacekm@af.czu.cz (M.P.); okrouhla@af.czu.cz (M.O.); gasparikm@af.czu.cz (M.G.); 2Department Animal Husbandry Science, Faculty of Agriculture, University of South Bohemia in České Budějovice, Studentská 1668, 370 05 České Budějovice, Czech Republic; jberan02@zf.jcu.cz

**Keywords:** body condition score, citric acid, Czech Fleckvieh cows, lipo-mobilisation, nutritional value

## Abstract

**Simple Summary:**

The intensity of negative energy balance period could potentially affect milk quality, mainly through its influence on milk composition. Fatty acid composition in milk affects its nutritional value for human consumption, because individual fatty acids have various effects on the human organism. The aim of our study was to describe the influence of negative energy balance on fatty acids proportion in milk of Czech Fleckvieh cows. Our results showed that cows in deep negative energy balance produced milk with a healthier proportion of fatty acids from a human nutrition standpoint. These cows experienced higher physiological stress after calving and are more susceptible to diseases and premature culling. Nowadays, farmers demand healthier and less problematic cows, while consumers ask for improved welfare and nutritionally balanced dairy products. However, using this phenomenon to improve nutritional value of milk might be opposed by consumers, farmers and even current breeding goals. These are currently focused on improving vitality, robustness, and longevity of dairy cows.

**Abstract:**

The objective of this study was to evaluate the influence of negative energy balance on fatty acids proportion in the milk of Czech Fleckvieh cows after calving. Milk quality was determined based on fatty acid group proportion. Milk quality was evaluated in relation to selected negative energy balance (NEB) traits: body condition change (DEC) and milk citric acid content (CAC) after calving. Milk samples (n = 992) were collected once per week from 248 Czech Fleckvieh cows during the first month of lactation. Fatty acid content (%) in milk samples was determined and results were grouped as saturated (SFA) (hypercholesterolemic or volatile fatty acids) or unsaturated (UFA) (monounsaturated or polyunsaturated). Our results showed that cows with a deep NEB produce milk that is healthier for human consumption. Cows with a more significant DEC or the highest level of CAC in milk had the lowest proportion of SFA and the highest proportion of UFA (*p <* 0.01). These cows experienced higher physiological stress after calving; however, they produced milk of higher nutritional quality. Nowadays, we can see preventive efforts to mitigate NEB periods as a result of modern breeding trends regarding vitality, robustness, or longevity.

## 1. Introduction

Cow milk is a rich source of several macro- and micronutrients [1] and is a fundamental dietary item in human nutrition [2]. Fat, as one of the main components of milk, has a huge influence on the taste and nutritional value of milk. Milk fat consists of individual fatty acids (FAs) and their triglycerides [3]. FAs in milk fat are usually divided into groups according to the length of their carbon chains and their saturation [4]. In particular, saturated (SFAs) and unsaturated FAs (UFAs) have recently been studied because of their important influence on human health [5]. The SFA group, especially the portion represented by the hypercholesterolemic FAs (HCFAs) [6], has been associated with increased deposition of fat in the vascular walls and increased risk of cardiovascular diseases [7]. Volatile fatty acids (VFAs) make up another part of the SFA group in milk. They are produced in the rumen and are indicative of the rumen’s fermentation intensity and utilisation efficiency of nutrients [8]. On the contrary, most UFAs are an important source for synthesis of biologically active substances, which improve human metabolism [9,10]. Monounsaturated fatty acids (MUFAs) [11] and polyunsaturated fatty acids (PUFAs) have been shown to have positive effects in the prevention of coronary heart and other diseases [12]. Conversely, some specific PUFAs such as trans-fatty acids could negatively impact human health [13].

The milk FA profile is affected by environmental factors [14,15,16,17], as well as genotype [18,19], and interactions between these factors. High-producing cows may face a state of negative energy balance (NEB) immediately after calving [20], which is manifested by utilisation of body fat reserves [21]. This is represented by a decline in the body condition score (BCS) [22], which is commonly used as a subjective indicator of energy balance. NEB intensity affects milk fat content [23] and the FA profile [24]. Metabolism products, such as urea [25], beta-hydroxybutyrate [26], and citric acid (CAC) [27], are created in the liver during this period. CAC in milk is easy to determine and can also be used as an indicator of NEB [28] or various diseases [29], primarily during the first month after calving [30]. Breeders try to counteract the negative influence of NEBs on cow health and the development of disease by environmental breeding adjustments, as well as through genetics. In fact, several selection indexes are focused on improving the wellness and increasing the robustness of Holstein and Jersey dairy cattle [31,32].

The family of Simmental breeds, which includes the Czech Fleckvieh, is the second largest dairy cattle population in central Europe after the Holstein cows. Although not as productive, their health, reproduction, and robustness traits are better than those of Holsteins [33]. Additionally, metabolism level differs between specialised milk breed (Holstein) and dual-purpose breed (Fleckvieh) [34]. Common goals in breeding programmes for Simmental cows are represented by their selection index (GZW), which has become increasingly focused on fitness, robustness and wellness traits [35]. Besides official indexes such as the GZW, various breeding companies focus their efforts on improving farms’ profitability and cows’ longevity by creation of special selection indexes or improved breeding values estimation for key traits, e.g., ketose, vitality [36], net merit index [37], wellness index [31] and others. Enhancing longevity by reducing early culling is not only a strategy to improve profit at dairy farms [38], but has also been related to improved animal welfare [39]. These breeding efforts are in accordance with farmers’ and consumers’ demands for healthier cows and better welfare. 

Period of NEB after calving could be partly considered as a welfare problem and partly as a metabolic one. In addition, the intensity of NEB could potentially affect milk quality, mainly through its influence on milk composition. Evaluation of milk samples for FA composition is a sophisticated way of determining milk quality for human health and nutrition. The aim of this study was to evaluate the influence of NEB on milk quality traits important for human diet, with a focus on FA composition, in Czech Fleckvieh cows. 

## 2. Materials and Methods 

### 2.1. Animals and Herd Management 

A total of 260 Czech Fleckvieh cows (73 in the first, 83 in the second, and 104 in the third and subsequent lactations) were included in this study. Tested animals had calved from October to April and did not have any previous reproductive or health disorders. The cows were housed in cubicles in a free-stall barn with straw bedding. Feeding rations were balanced in relation to daily milk yield and consisted of the same components throughout the evaluation period. All animals were fed a total mixed ratio (TMR) consisting of maize and alfalfa silage (21 kg per head day^−1^), alfalfa hay (10 kg per head day^−1^), brewery draff (5 kg per head day^−1^), bakery waste (1.5 kg per head day^−1^), straw (0.6 kg per head day^−1^), molasses (0.7 kg per head day^−1^), mineral supplements (0.1 kg per head day^−1^), and barley (from 2 to 10 kg per head day^−1^ based on feeding programme on farm). The feed during the experiment was optimised and physiologically balanced. In addition, straw intake from the bedding was not observed by us during the experiment nor farm staff in day to day operations. 

A body condition score (BCS) was used as a subjective NEB indicator. Body condition was scored using a 5-point scale with 0.25 point increments [40] at calving and 30 days after calving by one trained specialist. Declines in body condition between these two measurements were represented by body condition decrease (DEC). CAC in milk (mmol/L) was measured weekly and used as an objective NEB indicator [41]. Both indicators were used to divide tested animals into three groups, according to arithmetic means and standard deviation (< $\bar{x}$ -1/2s; -1/2s to 1/2s; > $\bar{x}$ + 1/2s). Therefore, for the DEC parameter, cows were divided into the following groups: 1. decrease by 1 point and more, 2. decrease by 0.75 to 0.5 points, 3. decrease by 0.25 points and less. Also, cows were divided based on CAC content into the following groups: 1. 10.37 mmol/L and less, 2. 10.37 – 12.6 mmol/L, 3. 12.60 mmol/L and more. 

### 2.2. Samples Collection and Analyses

Milk samples (n = 992) were collected weekly during the first 4 weeks of lactation. The first sample for each cow in the test was taken on the 7th day after calving. Aliquots of milk samples were collected without a preservative and in accordance with the milk recording system [42]. Milk samples were collected during standard milking procedure without any additional interference with the tested animal, therefore additional ethical approval was not required for the purpose of this study. This study was carried out in accordance with national legislation for protection of the animals against abuse (No. 246/1992) and with directive 2010/63/EU on the protection of animals used for scientific purposes. Milk samples were used for fat (F) extraction and fatty acid (FA) content. FA composition was extracted using the standard Röse-Gottlieb method (gravimetric) in accordance with EN ISO 1211 (ČSN 57 0534, 2010). The extract was obtained using a water-based-solution of ammonia, ethanol, diethylether and petrolether. FA methyl esters were prepared by the potassium hydroxide catalysed methylation and extracted into heptane. Gas chromatography of FA methyl esters was performed using the Master GC (DANI Instruments S.p.A.; Italy) (split regime, FID detector) on a column with polyethylene glycol stationary phase (FameWax – 30 mm × 0.32 mm × 0.25 μm). Helium was used as the carrier gas at a flow rate of 5 ml/min. The temperature settings used for gas chromatography (GC) was as follows: 50 °C (2 min), after which the temperature was increased to 230 °C at 10 °C/min (8 min), with the temperature of the detector being 220 °C.

Proportional composition of six FA groups (% SFA and its parts: % HCFA and % VFA; % UFA and its parts: % MUFA and % PUFA) was computed from 28 individual FA gravimetric contents (mg/100 g). The HCFA group consisted of lauric, myristic, and palmitic FA [43]. The VFA group consisted of butyric, caproic, caprylic, and capric FA [44]. UFA was taken into account as a positive group from a human nutrition point of view for further evaluation. 

The CAC was determined using the spectrophotometric method [41]. This analysis was done on spectrophotometer Genesys 10VIS (Thermo Scientific^TM^; Waltham; MA USA) with 428 nm wavelength. 

### 2.3. Statistical Analysis

The data were evaluated with the statistical software SAS 9.3. [45] using the UNIVARIATE, REG, CORR, GLM, and MIXED procedures. The GLM procedure was used to determine linear regressions between individual DEC respective to CAC and the proportion of FA groups (SFA, HCFA, VFA, UFA, MUFA, and PUFA) in milk. Linear regressions were corrected for the months of calving, lactation number, the repeated effect of animals, and regression based on days in milk (DIM). 

The REG procedure (STEPWISE option) was used for selection of suitable factors for the model equation. The best model for evaluation was selected in accordance with the values of the Akaike Information Criterion (AIC). The chosen model included the fixed effects of parity, groups of DEC or groups of CAC, month of calving, repeated random effect of the animal, and regression based on DIM. This model in MIXED procedure was used for evaluation of FA composition (SFA, HCFA, VFA; UFA, MUFA, and PUFA). For further evaluation, effects of DEC and CAC were divided into groups based on results of linear regression analysis. The effects of DEC and CAC were represented by three levels (≤ −1 point, −0.75 to −0.5 point, and ≥ −0.25 point; ≤ 10.37 mmol/L, 10.37 to 12.60 mmol/L, and ≥ 12.60 mmol/L, respectively). DEC levels indicated fat reserve loss during NEB, whereas CAC indicated the energy status of a cow, ranging from energy deficiency to energy abundance. The Tukey-Kramer method was used for evaluation of differences of least square means. The model equation was as follows:Y_ijkl_ = µ + PAR_i_ + MON_j_ + DEC_k_ + anim + b*(DIM) + e_ijkl_(1)
Y_ijkl_ = µ + PAR_i_ + MON_j_ + CAC_k_ + anim + b*(DIM) + e_ijkl_(2)
where: Y_ijkl_ = dependent variable (SFA, HCFA, VFA, UFA, PUFA, and MUFA in %); µ = mean value of dependent variable; PAR_i_ = fixed effect of i^th^ parity (i = the 1st lactation, n = 247; 2nd lactation, n = 299; 3rd and subsequent lactations, n = 446); MON_j_ = fixed effect of j^th^ month of calving (j = October, n = 96; November, n = 189; December, n = 191; January, n = 143; February, n = 145; March, n = 152; April, n = 76); DEC_k_ = fixed effect of k^th^ group of BCS decline (k = ≥ −1, n = 344; -0.75 to −0.5, n = 456; ≤ −0.25, n = 192); CAC_k_ = fixed effect of k^th^ group of citric acid content in milk (k = < 10.37 mmol/L, n = 278; 10.37 to 12.6 mmol/L, n = 376; > 12.60 mmol/L, n = 289); anim = repeated random effect of cow (248 cows); b*(DIM) = regression for days in milk during milk collection; e_ijkl_ = random error.

Significance levels of *p <* 0.05 and *p <* 0.01 were used to evaluate the differences between groups.

## 3. Results

Animals in our test achieved an average milk yield of 27.98 l with 4.75% fat during the first month of lactation. The average F content in milk decreased from 5.19% in the first week of lactation to 4.33% in the fourth week of lactation. On average, FA groups were SFA 70.73% (of which HCFA 41.22%, VFA 14.83%), MUFA 25.60%, and PUFA 3.64%. Body condition decreased (DEC) by 0.71 points on average (from 4.17 to 3.58 points) in the first month after calving. Maximal achieved DEC was −2.5 points and minimal observed DEC was +0.5 points. CAC was 11.62 mmol/L in the first week of lactation, 11.80 mmol/L in the second week of lactation, 11.34 mmol/L in the third week of lactation, and 11.20 mmol/L in the fourth week of lactation. 

Cows in a deeper NEB (≤ −1 point DEC) mobilised more fat reserves; therefore, they had more energy for metabolism and CAC was higher. Interestingly we only observed a weak negative correlation between our NEB indicators: DEC and CAC (r = −0.08; *p <* 0.05). Linear regression analysis showed interesting relationships between DEC and CAC and between SFA and UFA (R^2^ = 0.196, resp. 0.197). Each decrease of BCS by one point (DEC = -1) represented 3.53% decrease in SFA content (SFA = 70.52 + 3.53*DEC; *p* < 0.01) and an increase in UFA content by 3.53% (UFA = 29.47 − 3.53*DEC; *p <* 0.01). Each increase of CAC by one mmol/L represented a 0.60% decrease in SFA content (SFA = 76.20 − 0.60*CAC; *p* < 0.01) and an increase in UFA content by 0.59% (UFA = 23.80 + 0.59*CAC; *p <* 0.01). 

For our MIXED procedure, we used model equations that explained variability from 11.24% (VFA) to 21.87% (HCFA) and were statistically significant for all FA groups (*p <* 0.01). Effect of the DEC group was significant (*p <* 0.01) for all evaluated FA groups (Table 1); however, the effect of the CAC group was significant (*p <* 0.01) only for SFAs, HCFAs, UFAs, and MUFAs (Table 2). The effect of the month of calving was significant (*p <* 0.01) for all evaluated FA groups, whereas the effect of parity was not significant for HCFAs and VFAs. Finally, regressions based on DIM were significant (*p <* 0.05) for all FA groups, except VFAs. 

DEC, as our chosen subjective indicator of NEB, had a significant influence on FA composition (Table 1). The highest value for SFAs (+1.47 to +3.19%; *p <* 0.01) was determined for cows with the lowest DEC (≤ −0.25 points) during the first month of lactation. The same trends were also demonstrated for HCFA (+0.65 to +1.32; *p <* 0.05) and VFA (+1.10 to +2.47%; *p <* 0.01) content. In contrast, the lowest values of UFAs (−1.45 to −3.18%; *p <* 0.01) were observed for the same group of cows (DEC ≤ −0.25 points). We also observed the same trend for MUFAs (–1.25 to –2.82%; *p <* 0.05) and PUFAs (−0.21 to –0.36%; *p <* 0.05). 

Effects of CAC on FA composition (Table 2) had the opposite tendencies as the effects of DEC. We observed lower contents of SFAs (69.69%; *p <* 0.01) and HCFAs (40.40%; *p <* 0.01) for cows in the deepest NEB (CAC > 12.6 mmol/L = abundance of metabolic energy). Conversely, the content of “generally healthy” UFAs, MUFAs (*p <* 0.01) and PUFAs (*p* > 0.05) was increased (+1.08 to +2.71%, respectively +0.01 to +0.06%) for cows in this group.

Additionally, we observed higher content of SFAs (+2.35 to +4.07%), HCFAs (+0.59 to +0.88%), and VFAs (+0.60 to +1.10%) (*p <* 0.01) for cows on their first and second lactation compared to cows on third and subsequent lactations, whereas the results for UFAs (−2.35 to −4.07%), MUFAs (−2.13 to −3.72%), and PUFAs (−0.22 to −0.34%) were reversed. We also observed a higher proportion of UFAs, MUFAs, and PUFAs for animals calved from January to March compared to other months. 

## 4. Discussion

FAs in milk are currently the subject of numerous studies, mainly in respect to their variability in milk and importance to human nutrition. We observed similar contents of F, SFAs, MUFAs, and PUFAs in milk as did the authors of [46] in their experiment. BCS and its development reflects the amount of available energy in body fat and its utilisation [21,23]. On the contrary, CAC primarily reflects energy intake directly from the feed ratio and its utilisation. Therefore, DEC and CAC (r = −0.08; *p <* 0.05) are not interchangeable and both had significant effects on the FA groups in milk. The optimal physiological range for CAC in milk is 8 to 10 mmol/L [47]. Most of our tested cows were beyond this range, which indicated that they were in a metabolic state of energy surplus. This is typical for cows in NEB; therefore, it correctly corresponds with the condition decrease and increased milk production (from an average of 25.57 L in the first week to an average of 29.14 L in the fourth week of lactation) of tested cows. Cows in this period had accelerated metabolism and intensively utilised fat reserves [48]. CAC had a weak negative relationship to DEC, which was also indirectly observed in a study by the authors of [49]. CAC in milk could be used as an important descriptor of energy utilisation in cows. Naturally, part of the energy needed for the metabolic processes came from feed and the other part from body fat reserve utilisation. Therefore, cows with a high CAC probably utilise more energy from body fat, indicating a deeper NEB.

The importance of effects, such as lactation parity and month of calving, for milk analysis during NEB was also confirmed in other studies [15,50]. Season and phase of lactation affected milk composition, similar to that found in the studies by [51], [52], and [53]. This represented the farmers’ efforts to provide enough feed for dairy cows in early lactation to cover their energy needs. Deposited body fat is also reutilised during the period of intensive NEB [21], which is reflected by a change in FA composition in blood, and subsequently FA composition in milk. A deep NEB had a positive influence on milk nutritional quality, as was proven by our study. Milk of cows in the mildest NEB contained significantly higher amounts of SFAs in milk shown by [54]. Mild NEB is represented by a low body condition decline [55] and low CAC [47]. 

Interbreed differences in FA composition may exist, but the influence of NEB on the proportional composition of FAs was similar. Similar to that observed in the study by [56], Holstein cows in NEB also had lower SFA and higher UFA content. Our results also suggested that cows in a deeper NEB produce milk with a higher content of UFAs. In addition, Holstein cows in the study by [56] experienced similar DEC as our Czech Fleckvieh cows. On the contrary, the authors of [50] observed higher average DEC for Holstein cows compared to that of Simmental cows. HCFAs are precursors of cardiovascular diseases; therefore, it is important to focus on this FA group. Besides NEB, its content in milk can also be affected by feed supplements [7,57]. Our results indicated that the concentration of HCFAs was lower for cows with deeper NEB. However, the authors of [56] observed higher HCFA content for Holstein cows in deeper NEB. Apparently, the reaction of the organism to NEB is breed specific, and different metabolic pathways might be activated, resulting in milk with specific fatty acid proportions. Czech Fleckvieh cows are more robust and their metabolism might be more stable during critical periods such as deep NEB. Saving energy by not fully saturating fatty acids in milk during deep NEB might be crucial, because the organism has more energy to fully sustain its functions.

VFA content indicates rumen fermentation and effective utilisation of feed rations, which are nowadays considered as one of the most important breeding goals for the future of cattle farming [58]. The intensity of the NEB, represented by DEC and CAC, influenced the amount of VFAs in our study, as well as in the study by [59]. Therefore, cows with worse feed utilisation had lower contents of VFA and were in deep NEB. VFA content also increased during periods of positive energy balance, when CAC was low and there was only minimal utilisation of body energy reserves. 

On the contrary, the content of UFAs, including MUFAs and PUFAs, during this period decreased. Similarly, studies by [54] showed that cows with a greater decrease in body condition produced milk with a higher proportion of FAs beneficial to human nutrition. Consequently, FA composition in milk is more favourable during periods of intensive NEB. Milk with more than 12.60 mmol/L CAC could be considered to be most appropriate for human nutrition base of the concentration of UFA, MUFA, and PUFA groups. Similar relationships were observed by [56]. High CAC represented cows with a more intensive metabolism, which are able to receive more energy from feed intake and body fat reserve utilisation [47]. Therefore, CAC could be used as an indicator of milk nutritional quality.

It is evident that high-producing cows are not in their normal metabolic state during the first month of lactation. Our results suggested, that the NEB period could be used for the production of milk with a more favourable FA composition. Nevertheless, a deep NEB is not favourable for cow metabolism, metabolic robustness, health, welfare, and longevity as confirmed by [23], [60], and [36]. Breeding for “healthier” milk using this phenomenon is in contrast with modern selection trends for dairy cow as described in current studies [31,61,62]. Collection of raw milk from cows with a deep NEB in milk parlours is possible by creating fresh cow groups and subsequently separating their milk yield, or can be done automatically on farms with milking robots. This milk may be used to create dairy products enriched with specific FAs that have more favourable nutritional value.

Interestingly, the oldest cows produced nutritionally better milk from an FA standpoint. This is indirectly in accordance with worldwide breeding trends for dairy cows. Cow longevity and total economic profit is a substantial part of modern selection indexes [63]. Therefore, breeding for longevity could also produce better milk quality. Nevertheless, with continual parities of lactations, udder health is generally deteriorating and SCC in milk is usually increasing as well [64]. However, the problem of SCC is solvable with improved veterinary practices and technology of milk processing.

## 5. Conclusions

The aim of this study was to evaluate the influence of NEB and its subjective (BCS) and objective (CAC) indicators on milk quality traits from an FA composition standpoint in Czech Fleckvieh cows. Our results clearly showed that DEC and CAC have a negative relationship because each represents different aspects of the NEB period. Naturally, after using both indicators together, we gained a more complex picture about cows´ energy balance from an objective and subjective standpoint. Thus, we could correctly identify NEB intensity of tested cows for more precise evaluation of its influence on milk quality. Cows with a deeper NEB produced milk with higher proportion of FA (PUFA and MUFA) groups with benefits for human nutrition and health, whereas milk from animals with a mild NEB contained much higher SFAs. However, cows affected by deep NEB are at higher risk of being culled early in their production life. Our study shows that NEB indicators could be used as milk quality predictors or indirectly as cow wellness indicators within milk performance control systems. Moreover, besides economic and nutritional aspects, there are also environmental and ethical considerations that demand higher robustness and vitality to increase the productive life of dairy cows.

## Figures and Tables

**Table 1 animals-10-00563-t001:** Effects of body condition decline (DEC) groups on fatty acids group concentrations (LSM ± SELSM) and significance of effects in model equation.

	Groups of DEC			PAR	DEC	MON	b*(DIM)
	≤ −1 (n = 86 Animals with Four Repetition)	−0.75 to −0.5 (n = 114 Animals with Four Repetition)	≥ −0.25 (n = 48 Animals with Four Repetition)				
**SFA**	69.46 ± 0.30^C^	71.18 ± 0.25^B^	72.65 ± 0.40^A^	xx	xx	xx	xx
**HCFA**	40.81± 0.22^B,b^	41.48 ± 0.18^a^	42.13 ± 0.28^A^	xx	xx	xx	xx
**VFA**	13.40 ± 0.32^B^	14.77 ± 0.26^A^	15.87 ± 0.41^A^	NS	xx	xx	NS
**UFA**	30.51 ± 0.30^A^	28.78 ± 0.25^B^	27.33 ± 0.40^C^	xx	xx	xx	xx
**MUFA**	26.72 ± 0.29^A^	25.15 ± 0.24^B,a^	23.90 ± 0.38^B,b^	xx	xx	xx	xx
**PUFA**	3.79 ± 0.05^A,b^	3.64 ± 0.04^a^	3.43 ± 0.07^B,b^	xx	xx	xx	xx

SFA = saturated fatty acids; HCFA = hypercholesterolemic fatty acids; VFA = volatile fatty acids; UFA = unsaturated fatty acids; MUFA = monounsaturated fatty acids; PUFA = polyunsaturated fatty acids; PAR = parity; DEC = group of BCS decline; MON = month of calving; b*(DIM) **=** regression for days in milk during milk collection; xx = significant in model equation *p <* 0.01; NS = nonsignificant in model equation; Different letters within rows (A, B, C) indicate significant differences at the *p <* 0.01 level of significance; Different letters within rows (a, b) indicate significant differences at the *p <* 0.05 level of significance.

**Table 2 animals-10-00563-t002:** Effects of citric acid (CAC) groups on fatty acids group concentrations (LSM ± SELSM) and significance of effects in model equation.

	Groups of CAC			PAR	CAC	MON	b*(DIM)
	< 10.36 (n = 278)	10.36–12.60 (n = 376)	> 12.60 (n = 289)				
**SFA**	72,39 ± 0.34^A^	70,76 ± 0.30^A,a^	69.69 ± 0.34^B,b^	xx	xx	xx	x
**HCFA**	42.74 ± 0.24^A^	41.06 ± 0.20^B^	40.40 ± 0.24^C^	NS	xx	xx	xx
**VFA**	15.00 ± 0.35	14.44 ± 0.30	13.92 ± 0.35	NS	NS	xx	NS
**UFA**	27.58 ± 0.34^B^	29.22 ± 0.30^b^	30.27 ± 0.34^A,a^	xx	xx	xx	xx
**MUFA**	23.92 ± 0.32^C^	25.55 ± 0.28^B^	26.63 ± 0.33^A^	xx	xx	xx	x
**PUFA**	3.61 ± 0.06	3.66 ± 0.05	3.67 ± 0.06	xx	NS	xx	xx

SFA = saturated fatty acids; HCFA = hypercholesterolemic fatty acids; VFA = volatile fatty acids; UFA = unsaturated fatty acids; MUFA = monounsaturated fatty acids; PUFA = polyunsaturated fatty acids; PAR = parity; CAC = group of citric acid content in milk; MON = month of calving; b*(DIM) **=** regression for days in milk during milk collection; xx = significant in model equation *p <* 0.01; x = significant in model equation *p <* 0.05; NS = nonsignificant in model equation; Different letters within rows (A,B,C) indicate significant differences at the *p <* 0.01 level of significance; Different letters within rows (a, b) indicate significant differences at the *p <* 0.05 level of significance.

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
