# Peer review of "Negative Energy Balance Influences Nutritional Quality of Milk from Czech Fleckvieh Cows due Changes in Proportion of Fatty Acids"

_animals, 2020, doi:10.3390/ani10040563_

Round 1
Reviewer 1 Report
The authors have responded to all comments. Nevertheless, the conclusions of the study should be more clear and concise. After addressed, This manuscript is acceptable for publication.
Reviewer 2 Report
None
Reviewer 3 Report
I believe that the resubmitted manuscript is suitable for publication in Animals.
This manuscript is a resubmission of an earlier submission. The following is a list of the peer review reports and author responses from that submission.
Round 1
Reviewer 1 Report
The manuscript valuated the influence of negative energy balance on fatty acids proportion in milk of Simmental cattle after calving. Result explored that cows with a more significant DEC or the highest level of CAC in milk had the lowest proportion of SFA and the highest proportion of UFA that is healthier for human consumption. The findings are able to provide a new insight to use in milk quality prediction and cow wellness evaluation within milk recording systems. Despite the potential of the work, several issues will need further improvement.
1) Since this study clearly showed that both NEB indicators had significant relationships to FA groups. thuswise, what are the pathways and mechanisms of fatty acid metabolism in this process? It needs to be explained.
2) The purpose and prospects of this study need to be clearly presented.
3) Keywords, It is better to use Czech Fleckvieh cows or Simmental cows instead of dual purposed cows. "lipomobilisation" should be corrected as “ lipo-mobilisation”.
4) Detailed criteria for grouping lactating cows need to be given.
5) Table 1, what's the number of individuals in each group?
6) Why "n6:n3" was ignored in the analysis of unsaturated fatty acids in milk?
7) There are numerous grammars and tense errors in the article. Please revise all the manuscript carefully.
Author Response
Dear reviewers,
First of all, I wanted to thank you for the work you put into improving our paper. Thanks to your recommendations, we manage to improve the article. We appreciate all the hard work you have put into improving this paper. We also want to apologize for all the glaring mistakes we didn’t noticed before submitting the manuscript.
Most of the proposed revisions were applied, but few of them we couldn’t change. Below, we answer to all the comments one by one (answers are bold italic). We hope that you will be satisfied with revisions we made after your comments, and also satisfied with the quality of this paper.
Specific comments:
1) Since this study clearly showed that both NEB indicators had significant relationships to FA groups. Thus wise, what are the pathways and mechanisms of fatty acid metabolism in this process? It needs to be explained.
Thank you for your recommendation to include more detailed biochemical pathways and mechanisms of fatty acids metabolism. We know, that depot fat is intensively mobilized and destroyed on glycerol and fatty acids during NEB period. Some of these fatty acids are directly transferred to milk, although some are not. Nevertheless, this article is primarily focused on phenotypical manifestation, therefore biochemical or detail physiological explanation of this process are out of our field of interest or knowledge.
2) The purpose and prospects of this study need to be clearly presented.
Thank you for comment. The aim of this study was more specified, and purpose was reformulated (L87 – 89).
3) Keywords, It is better to use Czech Fleckvieh cows or Simmental cows instead of dual purposed cows. "lipomobilisation" should be corrected as “ lipo-mobilisation”.
Thank you. The keywords were changed based on your recommendation. In addition, “lipomobilisation” was changed for “lipo-mobilisation” in whole article.
4) Detailed criteria for grouping lactating cows need to be given.
Thank you for your comment. We specified group creation in methodology for DEC and CAC parameters as follows: “Therefore, for DEC parameter cows were divided into following groups: 1. decrease by 1 points and more, 2. decrease by 0.75 to 0.5 points, 3. decrease by 0.25 points and less. Also, cows were divided based on CAC content into following groups: 1. 10.37 mmol/l and less, 2. 10.37 – 12.6 mmol/l, 3. 12.60 mmol/l and more.”(L111-115).
5) Table 1, what's the number of individuals in each group?
Thank you for recommendation. The numbers of individual animals in each group were added to the table 1.
6) Why "n6:n3" was ignored in the analysis of unsaturated fatty acids in milk?
Yes, thank you for this question. We agree that n6:n3 ratio is very important and should be discussed in length. However, after further discussion with my team, we decided to focus on those results in more detail in future manuscripts, which are currently under preparation.
7) There are numerous grammars and tense errors in the article. Please revise all the manuscript carefully.
Yes we apologize for the grammar mistakes. Article was again re-read by me and fluent English speaker, and most of the mistakes were corrected. Manuscript was proof read by native speaker. Elsevier corrected – project nr. 168517.

Reviewer 2 Report
Animals 708490
General comment
In the Introduction I have missed some crucial points. First of all, Simmental are classified as dual-purpose animals, therefore one would expect the intensity and incidence of NEB to be differ compared to Holstein. Please add this comparison and support it with some reference.
However, the main issues are related to n. of herds (1) of the study and to the period considered. Based on M&M, it seems that summer calving cows are missing. I expect changes in forage quality, herd management and climatic conditions, that could have a strong impact on variability of trait considered. Authors should know very well that this may be an important issue and should properly justify this lack, since the focus in on the first month of lactation and month of calving (i.e. season) is here the key factor. Pay attention to speculations made in the conclusions; milk quality has several interpretations.
From L149, it is not clear if cow was included in the model as random or fixed effect.
Specific comments:
L50: ‘diseases’ instead of ‘disease’
L56-57: need to be rephrased
L58: ‘Milk FA profile’
L59: not all cows manifest NEB. Please change to ‘High-producing cows may face up a state of…’
L60: BCS estimates the degree of lipid accumulation and is useful before parturition. Not sure if BCS is also routinely monitored for NEB detection in the early lactation.
L66: ‘Breeders try to counteract ’
L67: ‘as well as through genetics. In fact, several selection indexes are focused …’. Please add reference and the breeds you are referring to.
L71: ‘Holstein cows’.
L80-85: rephrase like ‘The aim of this study was to evaluate the influence of NEB on milk quality traits important for human diet, including FA composition, in Czech Fleckvieh cows.’
L98: information on the evaluator(s) is missing
L113-120: Add details of the methodologies used for fat extraction and CAC determination.
L122: when was MIXED procedure used?
L129-130: Delete. Breed has only 1 level, as well as year (of sampling or of calving?). Specify that each cow was sampled only in a single lactation.
L144 and L174: delete ‘of lactation’
L154 and L206: Define ‘F’
L155: ‘On average, FA groups were..’
L157: ‘in the first month after calving’ instead of ‘during tested period’
L159-160: are you talking about (raw) means or LSM here? Please provide the 4 averages CAC of the 4 samplings.
L161: ‘mobilized’ instead of ‘utilised’. This sentence needs a reference.
L163: ‘between DEC and CAC and between SFA and UFA’
L177: delete ‘significantly’ as you provide P value
L199-203: better to move in Discussion
L209-210: provide coefficient of correlation for DEC and CAC from your data to support your statement
L211: the study [46] is based on bulk milk, that is usually less variable than individual milk
L212-214: it may be helpful to include info on milk production of your cows to support this concept
L258: ‘high-producing cows’
L261: welfare
L263-264: it is easy if you adopt management strategies as milking groups (‘fresh’ cows milked separately)
L265-268: unclear; do old cows in early lactation produce the best milk from FA point of view? But what about SCC (udder health indicator that increase with parity order)?
Conclusions: you might briefly specify your aim and your choices (e.g. why DEC and CAC as NEB indicator). Too strong assumptions; consider that cows with marked NEB may not have good longevity and are at greater risk to be culled early in productive life. Also, define to which ‘milk quality’ you are referring to (L267-268), e.g. the milk technological quality (cheese manufacture) is not better in multiparous if we consider SCC level and acidity of milk.
Table 1.
- I am confused in the interpretation of letters used for LSM comparisons. Why some LSM have 2 equal letters (‘Aa’ or ‘Bb’)?
- In the caption: is “among rows” correct? Should not be “within rows”?
- Then, consider that usually you have to start in alphabetical order from the greatest LSM (e.g. for SFA: 72.65 should be ‘A’ and 69.46 should be ‘C’).
Both tables. PAR, MON and DIM may be omitted
Author Response
Dear reviewers,
First of all, I wanted to thank you for the work you put into improving our paper. Thanks to your recommendations, we manage to improve the article. We appreciate all the hard work you have put into improving this paper. We also want to apologize for all the glaring mistakes we didn’t noticed before submitting the manuscript.
Most of the proposed revisions were applied, but few of them we couldn’t change. Below, we answer to all the comments one by one (answers are bold italic). We hope that you will be satisfied with revisions we made after your comments, and also satisfied with the quality of this paper.
Specific comments:
In the Introduction I have missed some crucial points. First of all, Simmental are classified as dual-purpose animals, therefore one would expect the intensity and incidence of NEB to be differ compared to Holstein. Please add this comparison and support it with some reference.
Thank you for your comment. We are aware of this fact, and therefore we added citation (34) describing differences between Holstein and Fleckvieh metabolic status (L73-75).
However, the main issues are related to n. of herds (1) of the study and to the period considered. Based on M&M, it seems that summer calving cows are missing.
Thank you for your comment. Heat stress is manifested in summer period (more diseases, low milk production). For this reason our monitoring was realised out of this period. Also capacity of observation was limited on 1000 analysis and we decided to focus on limited period.
From L149, it is not clear if cow was included in the model as random or fixed effect.
We apologize for this mistake. Cow was included in the model as random effect. This mistake was corrected in text L140, L157-158.
L50: ‘diseases’ instead of ‘disease’
We apologize for this mistake. It was rewritten.
L56-57: need to be rephrased
Thank you for your recommendation. This sentence was reformulated: “Contrary, some specific PUFAs as trans-fatty acids could negatively impact human health.”
L58: ‘Milk FA profile’
Thanks you for your correction. This sentence was corrected based on your recommendation (L58).
L59: not all cows manifest NEB. Please change to ‘High-producing cows may face up a state of…’
We apologize for this mistake. This sentence was corrected based on your recommendation.
L60: BCS estimates the degree of lipid accumulation and is useful before parturition. Not sure if BCS is also routinely monitored for NEB detection in the early lactation.
Thank you for your comment. BCS is the cheapest and most common method of nutrition status monitoring and management in Czech farms. Moreover, this method can be used for the objective measurements as well, for example DELAVAL monitoring system for BCS evaluation. Increase of data collection is supported by new trends of precision agriculture.
L66: ‘Breeders try to counteract ’
Thank you for your recommendation. The sentence on L66 was rewritten according to recommendation.
L67: ‘as well as through genetics. In fact, several selection indexes are focused …’. Please add reference and the breeds you are referring to.
Thank you for comment. This sentence was corrected based on your recommendation and citation was added.
L71: ‘Holstein cows’.
Thanks you for your correction. The word “cows” was added to the sentence (L72).
L80-85: rephrase like ‘The aim of this study was to evaluate the influence of NEB on milk quality traits important for human diet, including FA composition, in Czech Fleckvieh cows.’
Thank you for your recommendation. The text on line 87-89 was corrected and added recommended sentence: “The aim of this study was to evaluate the influence of NEB on milk quality traits important for human diet, including FA composition, in Czech Fleckvieh cows.“
L98: information on the evaluator(s) is missing
We apologize for this mistake. Information on evaluator was added. Also, “by one trained specialist” was added to sentence (L107-108).
L113-120: Add details of the methodologies used for fat extraction and CAC determination.
Thank you for your recommendation. Following sentence was added to the chapter materials and methods as explanation of FA determination: “FA composition was extracted using the standard Röss-Gottlieb method (gravimetric) in accordance with EN ISO 1211 (ČSN 57 0534, 2010). The extract was obtained using a water-based-solution of ammonia, ethanol, diethylether and petrolether. FA methyl esters were prepared by the potassium hydroxide catalysed methylation and extracted into heptane. Gas chromatography of FA methyl esters was performed using the Master GC (DANI Instruments S.p.A.; Italy) (split regime, FID detector) on a column with polyethylene glycol stationary phase (FameWax – 30 mm x 0.32 mm x 0.25 μm). Helium was used as the carrier gas at a flow rate of 5 ml/min. The temperature settings used for GC was as follows: 50 °C (2 min), after which the temperature was increased to 230 °C at 10 °C/min (8 min), the temperature of the detector being 220 °C.”(L126-135).
Following text was added as explanation of CAC determination: “This analysis was done on spectrophotometer Genesys 10VIS (Thermo ScientificTM; USA) with 428 nm wavelength.”(L141-142).
L122: when was MIXED procedure used?
Thank you for comment. The MIXED procedure was used for finally evaluation. This is now explained on L155.
L129-130: Delete. Breed has only 1 level, as well as year (of sampling or of calving?). Specify that each cow was sampled only in a single lactation.
Thank you for your recommendation. The sentence: “Effects of breed and year were eliminated from the model equation because of uniform environmental conditions.” was deleted from the text.
L144 and L174: delete ‘of lactation’
Thanks you for your correction. This sentence was corrected based on your recommendation.
L154 and L206: Define ‘F’
We apologize for this mistake. F (fat) was defined at first using in text.
L155: ‘On average, FA groups were.’
Thank you for your recommendation. This sentence was reformulated according your recommendation (L178).
L157: ‘in the first month after calving’ instead of ‘during tested period’
Thank you for your recommendation. This sentence was reformulated according your recommendation (L180-181).
L159-160: are you talking about (raw) means or LSM here? Please provide the 4 averages CAC of the 4 samplings.
Thank you for comment. Requested information was added to text.
L161: ‘mobilized’ instead of ‘utilised’. This sentence needs a reference.
Thanks you for your correction. This sentence was corrected based on your recommendation. Reference for this sentence cannot be added to “Results” text, but this problem is explained in “Introduction”, and further in “Discussion” section with references 21, 22 and 23.
L163: ‘between DEC and CAC and between SFA and UFA’
Thanks you for your recommendation. This sentence was corrected based on your recommendation (L189).
L177: delete ‘significantly’ as you provide P value
Thanks you for your correction. This sentence was corrected based on your recommendation.
L199-203: better to move in Discussion
Thank you for recommendation. After further discussion with the team we decided to reformulated these sentences, but keep them in the results. In discussion we want to discuss specific results that were focus of this experiment.
L209-210: provide coefficient of correlation for DEC and CAC from your data to support your statement
Thank you for your comment. Correlation coefficient was added to the text „(r = -0.08; P < 0.05)” (L237).
L211: the study [46] is based on bulk milk that is usually less variable than individual milk
Thank you for comment. The author's team is aware of the differences in citric acid content in bulk and individual milk samples. Metabolism of each cow is different, and we can observed certain level of individualism in metabolic intensity, resp. different citric acid content. Mentioned citation was used to set boundaries and standard levels to make it easier.
L212-214: it may be helpful to include info on milk production of your cows to support this concept
Thank you for your recommendation. Information about increase of milk production (from average of 25.57 l in the first week to average of 29.14 in the four week of lactation) was added to text (L241-243).
L258: ‘high-producing cows’
Thanks you for your recommendation. This sentence was corrected based on your recommendation (L288).
L261: welfare
Thanks you for your recommendation. This word was added to text (L2¨91).
L263-264: it is easy if you adopt management strategies as milking groups (‘fresh’ cows milked separately)
Thank you for your recommendation. This option is outlined in discussion now (L293-297).
L265-268: unclear; do old cows in early lactation produce the best milk from FA point of view? But what about SCC (udder health indicator that increase with parity order)?
Thank you for your comment. Following sentence with citation was added to discussion: “Nevertheless with continual parties of lactations, udder health is generally deteriorating and SCC in milk is usually increasing as well [64]. However, problem of SCC is solvable with improved veterinary practices and technology of milk processing.”(L301-303).
Conclusions: you might briefly specify your aim and your choices (e.g. why DEC and CAC as NEB indicator). Too strong assumptions; consider that cows with marked NEB may not have good longevity and are at greater risk to be culled early in productive life. Also, define to which ‘milk quality’ you are referring to (L267-268), e.g. the milk technological quality (cheese manufacture) is not better in multiparous if we consider SCC level and acidity of milk.
Thank you for your comment. The chapter “Conclusion” was reformulated and simplified. Briefly stated aim of our manuscript was added to the beginning of this section (L305-306).
Table 1.
- I am confused in the interpretation of letters used for LSM comparisons. Why some LSM have 2 equal letters (‘Aa’ or ‘Bb’)?
- In the caption: is “among rows” correct? Should not be “within rows”?
- Then, consider that usually you have to start in alphabetical order from the greatest LSM (e.g. for SFA: 72.65 should be ‘A’ and 69.46 should be ‘C’).
We apologize for this mistake. Tables were checked and modified according to your recommendations.
Both tables. PAR, MON and DIM may be omitted
Thank you for your comment. We believe, that this information in table 1 and 2 are important from the overall model characterization, as they are informative in the wider context of expressed relations. After further discussion with the team we would rather left PAR, MON and DIM in the tables.

Reviewer 3 Report
It is nice that the study was conducted on a different breed that Holstein. However, some caution should be exercised when reproducing the results. Czech Fleckvieh is dual purpose breed with a lower yield than Holstein.
Some comments:
line 91 - it is not clear if the cows were housed in cubicles?
line 92 - straw bedding allows for a certain straw intake in addition to the feed that is in TMR.
Author Response
Dear reviewers,
First of all, I wanted to thank you for the work you put into improving our paper. Thanks to your recommendations, we manage to improve the article. We appreciate all the hard work you have put into improving this paper. We also want to apologize for all the glaring mistakes we didn’t noticed before submitting the manuscript.
Most of the proposed revisions were applied, but few of them we couldn’t change. Below, we answer to all the comments one by one (answers are bold italic). We hope that you will be satisfied with revisions we made after your comments, and also satisfied with the quality of this paper.
Specific comments:
- line 91 - it is not clear if the cows were housed in cubicles?
Thank you for noticing this missing information. Cubical system is very dominant in Czech Republic, so we forgot to add this information. However, we agree it is important for experimental repetition; and therefore added in the “Materials and Methods” section (L97-98).
- line 92 - straw bedding allows for a certain straw intake in addition to the feed that is in TMR.
Thank you for interesting comment. Yes, it’s true that cows can intake additional straw from bedding although this is mostly occurs during metabolic problems, when the fibre is missing from the feed. Following sentence was added to the chapter materials and methods “The feed during the experiment was optimized and physiologically balanced. In addition, straw intake from the bedding was not observed by us during experiment nor farm staff in day to day operations.”(L103-105).

Reviewer 4 Report
Presented data have a certain value because they contribute to the understanding a topic relevant to the dairy sector, although overall they do not add information of great novelty compared to what is already known or expected on the basis of what is known about and physiology and metabolism of dairy cows. However, the paper has many weaknesses regarding contextualization (introduction), methods, presentation and discussion of the results. Here are some of them.
The chemical composition and nutritional value of the diet fed under experimentation are not reported; this makes replication of the experiment really difficult; the simple proportion of single feeds is not enough to characterize the diet which could significantly affect the acidic profile of milk according its content in fiber fractions, energy and protein content, energy and protein ratio, digestibility of starches etc. ..
The characteristics of the acidic profile of milk with reference to its effects on nutrition and human health have been oversimplified by considering only the classification in saturated(HCFA and VFA) and unsaturated (MUFA and PUFA).
The number of cows in different groups is not reported.
The statistical analysis used seems not to be presented correctly and in any case not fully justified or lacking, for example the outcome of a multicollinearity test on the independent variables is not reported which, considering the acidic profiles of the milk, would be most appropriate.
The discussion of the results is inadequate; the discussion and conclusions include arguments that do not seem justified by the focus and results of the study.

Author Response
Dear reviewers,
First of all, I wanted to thank you for the work you put into improving our paper. Thanks to your recommendations, we manage to improve the article. We appreciate all the hard work you have put into improving this paper. We also want to apologize for all the glaring mistakes we didn’t noticed before submitting the manuscript.
Most of the proposed revisions were applied, but few of them we couldn’t change. Below, we answer to all the comments one by one (answers are bold italic). We hope that you will be satisfied with revisions we made after your comments, and also satisfied with the quality of this paper.
Specific comments:
The chemical composition and nutritional value of the diet fed under experimentation are not reported; this makes replication of the experiment really difficult; the simple proportion of single feeds is not enough to characterize the diet which could significantly affect the acidic profile of milk according its content in fiber fractions, energy and protein content, energy and protein ratio, digestibility of starches etc. ..
Thank you very much for your comment. We agree with the principle of commentary, however, the chemical composition and/or nutritional value of feed ration were not evaluated, resp. examined in detail within manuscript. The feed ration was optimized and balanced for concrete physiologically requirements of breed during specific stage of lactation (L99-105). The main task of study was to monitor and evaluate changes of dependent variables (SFA, UFA, etc.) in accordance with intensity of negative energy balance under field conditions in standard Czech Fleckvieh dairy farm operated within the Czech Republic. The course of energy balance intensity in Fleckvieh cattle during lactation differs comparing with Holstein. Therefore, authors tried to detect potential differences in Czech Fleckvieh as a national genetic resource and to evaluate results of its milk nutritional quality from that point of view at the same time. Presented manuscript is a part of results based on wider practical monitoring and is focusing on relationship among changes of energy balance indicators and composition of FA groups only. Authors considered remark of reviewer very carefully and assume information level of manuscript is corresponding from the points of view mentioned above.
The characteristics of the acidic profile of milk with reference to its effects on nutrition and human health have been oversimplified by considering only the classification in saturated (HCFA and VFA) and unsaturated (MUFA and PUFA).
The aim of this study was evaluated effect of NEB indicators changes on proportion of main groups of FA and selected specific fatty acids according to manuscript published within that area previously. Authors agree with reviewer its simple view, however it wasn´t postulated to evaluated proportion of all FAs in milk. The main goal of observation and evaluation was to determine objective differences in FA groups in relation to the course of energy balance during the 1st phase of lactation. Current direction of dairy cows’ selection and breeding focusing on mitigation of NEB can bring benefits for production and economic parameters as well as for animal welfare. However, changes in FA groups composition document this direction will bring changes in milk composition as well, lower nutritional quality of milk (higher SFA, higher HCFA etc.). This is the main goal of manuscript, to emphasize that point of view and, maybe, to provoke discussion on this topic, because consumers are observing all aspects of farm animal breeding more and more.
The number of cows in different groups is not reported.
Thank you for your comment. The numbers of individual animals in each group were added to the table 1. The numbers of observation was corrected in full text.
The statistical analysis used seems not to be presented correctly and in any case not fully justified or lacking, for example the outcome of a multicollinearity test on the independent variables is not reported which, considering the acidic profiles of the milk, would be most appropriate.
We thank you for your feedback, but we believe that the chosen statistical procedures are absolutely adequate and correct for the purposes of this evaluation. The results were continuously verified by means of partial statistical evaluations, which were not published for clarity. Effects in equation modes were available to be targeted based on standard methods for selecting available models (REG STEPWISE).
The discussion of the results is inadequate; the discussion and conclusions include arguments that do not seem justified by the focus and results of the study.
These sections were reworked fundamentally. Changes are based on wide recurrent discussion of the authors’ team according to the specific requirements of other opponents.
